# Trees Diversity and Species with High Ecological Importance for a Resilient Urban Area: Evidence from Cotonou City (West Africa)

Assouhan Jonas Atchadé [1,2,*], Madjouma Kanda [2], Fousseni Folega [2], Hounnankpon Yédomonhan [3], Marra Dourma [2], Kperkouma Wala [2] and Koffi Akpagana [2]

1 Regional Center of Excellence on Sustainable Cities in Africa (CERViDA), University of Lomé, Lomé BP 1919, Togo
2 Laboratory of Botany and Plant Ecology (LBEV), Faculty of Science, University of Lomé, Lomé BP 1919, Togo
3 Botany and Plant Ecology Laboratory, Faculty of Science and Technology, University of Abomey-Calavi, Cotonou 01 BP 4521, Benin
* Correspondence: assouhan.atchade@cervida-togo.org

**Abstract:** Rapid urbanization and climate change effects may cause dramatic changes in ecosystem functions in cities, thereby inevitably affecting the growth performance of old trees. Few studies have explored species diversity and spatial differentiation in Benin urban areas. This study aims to explore this dimension of urban ecology in order to build resilience to climate change in the city of Cotonou. Its objective was to determine the predominant level of tree diversity in the city's land use units. The urban green frame was subdivided into six land use units, namely, establishments, residences, green spaces, commercial areas, administrative areas, and roads. The forest inventories were carried out in 149 plots with surfaces evaluated at 2500 m$^2$ each. The IVI, an index that highlights the relative density, relative dominance, and relative frequency of species, has been used to characterize the place occupied by each species in relation to all species in urban ecosystems. This shows ecological importance through the diversity and quality of ecosystems, communities, and species. A total of 62 tree species in 55 genera and 27 families were recorded. The results show that the flora of the city of Cotonou is characterized by a strong preponderance of exotic species with some differences in species presence. The most abundant species with high ecological importance (IVI) in the different types of land use of the city are *Terminalia catappa* (IVI = 121.47%), *Terminalia mantaly* (IVI = 90.50%), *Mangifera indica* (IVI = 64.06%), and *Khaya senegalensis* (IVI = 151.16%). As the use of ecosystem services is recommended to tackle urban climate hazards, this study shows that direct development of this urban vegetation could improve the resilience of urban life to climate hazards through the provision of urban ecosystem services, potential ecological infrastructure foundations, and urban nature-based solutions.

**Keywords:** urban tree importance; diversity indexes; ecosystem services; species abundance; urban ecological planning; conservation planning; ecological value index; Cotonou



## 1. Introduction

We have entered the Anthropocene—an era where humans are a dominant geological and ecological force—and at the same time, we have entered an urban era. More than half of humanity now lives in cities, and by 2030, this proportion will reach 60% [1]. In other words, in just over two decades, from 2010 to 2030, an additional 1.5 billion people will be added to the urban population. Creating healthy, livable urban spaces for so many additional people will be one of the major challenges of our time. The quality of the urban environment, both its built and natural components, will determine the quality of life for an estimated total of five billion people, existing and future, by 2030 [2]. In the same vein, scientists have begun by questioning the place of the urban environment and its spaces with

a natural character (known as urban vegetated spaces) in the protection and enhancement of biodiversity and optimization of ecosystem services [3].

In Benin, a West African country, the urbanization rate increased from 11% in 1960 to 40% in 1990, and then from 42% in 2005 to 44% in 2015 [4]. Moreover, in future projections, more than half of Benin's population will live in cities by 2025, with an estimated urban population rate of 56.2%. Urban anthropogenic activities have enormous impacts on biodiversity and ecosystem services [5,6]. Simultaneously balancing the need for urban growth with biological conservation at a reasonable threshold remains a concern in an approach to perpetually improving the quality of the living environment in high-concentration areas [7–9]. In urban areas with current environmental issues like climate change and biodiversity loss, trees contribute to air purification through carbon dioxide sequestration, microclimate creation, urban tree species restoration, and land attractiveness [10,11]. Atchadé et al. [12] have demonstrated through stakeholders' consultation that urban trees in Cotonou City have the ability to mitigate urban heat islands and stormwater and strengthen human health. These benefits attribute social, economic, and environmental functions to trees [13,14]; the importance of which is reflected in ecosystem services. Although urban floral diversity is essential for providing ecosystem services and enhancing human well-being, it remains threatened by the anthropogenic and environmental consequences of urbanization [15–17].

According to Folega et al. [9] and Alpaidze et al. [18], it is evident that urban nature planning, management, and development can be combined with biodiversity conservation and preservation objectives for the perpetual improvement of the living environment in a coherent urban territory. The literature reveals that little research has been conducted on the composition and structure of urban forests in urban centers in West Africa, particularly in cities with high population growth (Nero et al. [17]). In Benin, the studies of Sehoun et al. [19] focused on the study of species diversity in green spaces in the city of Cotonou; the studies of Teka et al. [20] addressed the effect of the avenue trees of the Boulevard of Missèbo-Zongo on the local microclimate of the city of Cotonou, while Orou et al. [21] assessed the diversity and structure of woody vegetation in the city of Malanville in North Benin. Although research has been conducted based on species diversity in the city [22,23], the distribution and prioritization of ecologically important species in urban land use units is lacking. The lack of urban biodiversity data in tropical cities and the immediate need to undertake research within the current frontiers of urban ecology are highlighted by [24]. The ecological information deficit could be one of the major hindrances to the design of national urban forestry and environmental improvement programs, and it could also hinder the optimization of ecosystem services needed to circumvent and adapt to the effects of climate change in a sustainable urban development context. Therefore, the present work aims to fill these knowledge gaps by

-   evaluating the floristic diversity in terms of tree composition of the different land use units through different ecological indexes.
-   prioritizing the top species that have ecological importance in order to allow future studies related to climate impacts on them and their preservation.
-   making recommendations to protect urban species from climate change impacts and other threats (like species invasion).

The results could help in guiding the green development policies of future African and Beninese cities.

## 2. Methodology

### 2.1. Study Area

The city of Cotonou is in the south of the Republic of Benin between 6°20′ and 6°23′ north latitude and 2°22′ and 2°30′ east longitude. It is bordered to the north by Lake Nokoué, to the west by the Commune of Abomey-Calavi, to the east by the Commune of Sèmè-Kpodji, and to the south by the Atlantic Ocean (Figure 1). The city covers an area of 79 km$^2$ [23].

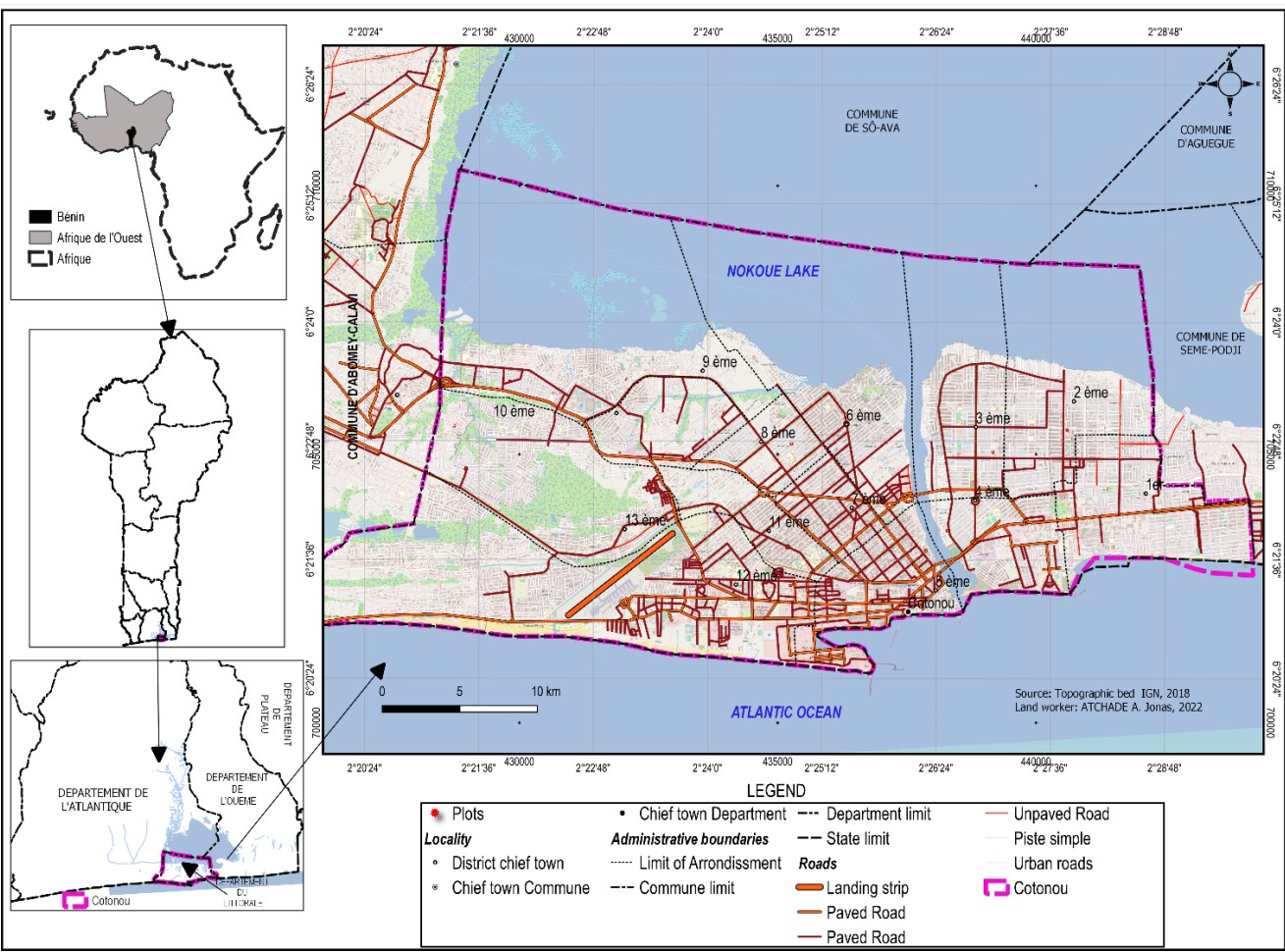

**Figure 1.** Location map of the research area.

Administratively, the city of Cotonou comprises 13 arrondissements subdivided into 144 neighborhoods. Its population is 679,012 inhabitants, according to the general population and housing census [4]. The climate is humid subequatorial, with two dry seasons (mid-November to mid-March and mid-July to August) and two rainy seasons (mid-March to mid-July and September to mid-November). The average annual rainfall is 1200 mm, with 700–800 mm in the long rainy season and 400–500 mm in the short rainy season [25]. The average temperature in the coastal zone is 26.8 °C, with extremes of 16.5 °C and 36.6 °C. The average relative humidity in Cotonou is 84%. The hydrographic network consists of Lake Nokoué and the Atlantic Ocean. The types of soil encountered include sandy soils, ferruginous soils, and hydromorphic soils [19]. All these characteristics favor plant development. The current urban matrix of the city offers a wide range of types of artificial and natural environments and vegetation ranging from totally unvegetated environments in the city centers to wooded private parks in residential areas to spontaneous vegetation in abandoned estates in the neighborhoods to fallows, plantations, ponds, marshes, and swamps in the peripheral areas of the city [26].

## 2.2. Data Collection and Analysis

After having mapped the city of Cotonou, we randomly selected 8 of the 13 boroughs in the city to cover all the localities (central and peripheral boroughs) as recommended by Moussa et al. [21] and Amontcha [27]. This sampling approach follows the scientific logic used by Moussa et al. [21] and Foléga et al. [9] in the cities of Niamey (Niger) and Atakpamé (Togo), respectively. These countries border Benin.

Based on FAO's [26] definition of an urban forest, the study area was stratified into six land use types that correspond to urban forests in Cotonou: (1) commercial areas including markets, stores, boutiques, restaurants, and vehicle repair shops; (2) roads covering main streets and boulevards; (3) residential areas covering houses, mosques, and churches; (4) schools covering private and public training and learning institutions such as elementary school, secondary schools, universities, schools, and vocational training centers; (5) administrative areas such as public and private utilities; and (6) urban green space areas consisting of urban agricultural plots, agroforestry systems, wetlands, irrigated agricultural land, and botanical gardens. In view of the type of land use, rectangular and square plots of 25 m × 100 m and 50 m × 50 m, respectively, were set up (Figure 2). This approach to plot sizes follows what was recommended by Thiombiano et al. [28] to harmonize forest sampling in the West Africa area. As the shape of the roadway is purely different from that of other occupancy units, such as administrative and the like, linear plots were installed along the tracks. In each of the plots, floristic surveys collected data on all trees with a diameter at breast height (DBH) ≥ 10 cm. Data for each tree included scientific name, family, and genera, as well as DBH, height, and number of stems. Data collection ran from May 2022 to October 2022.

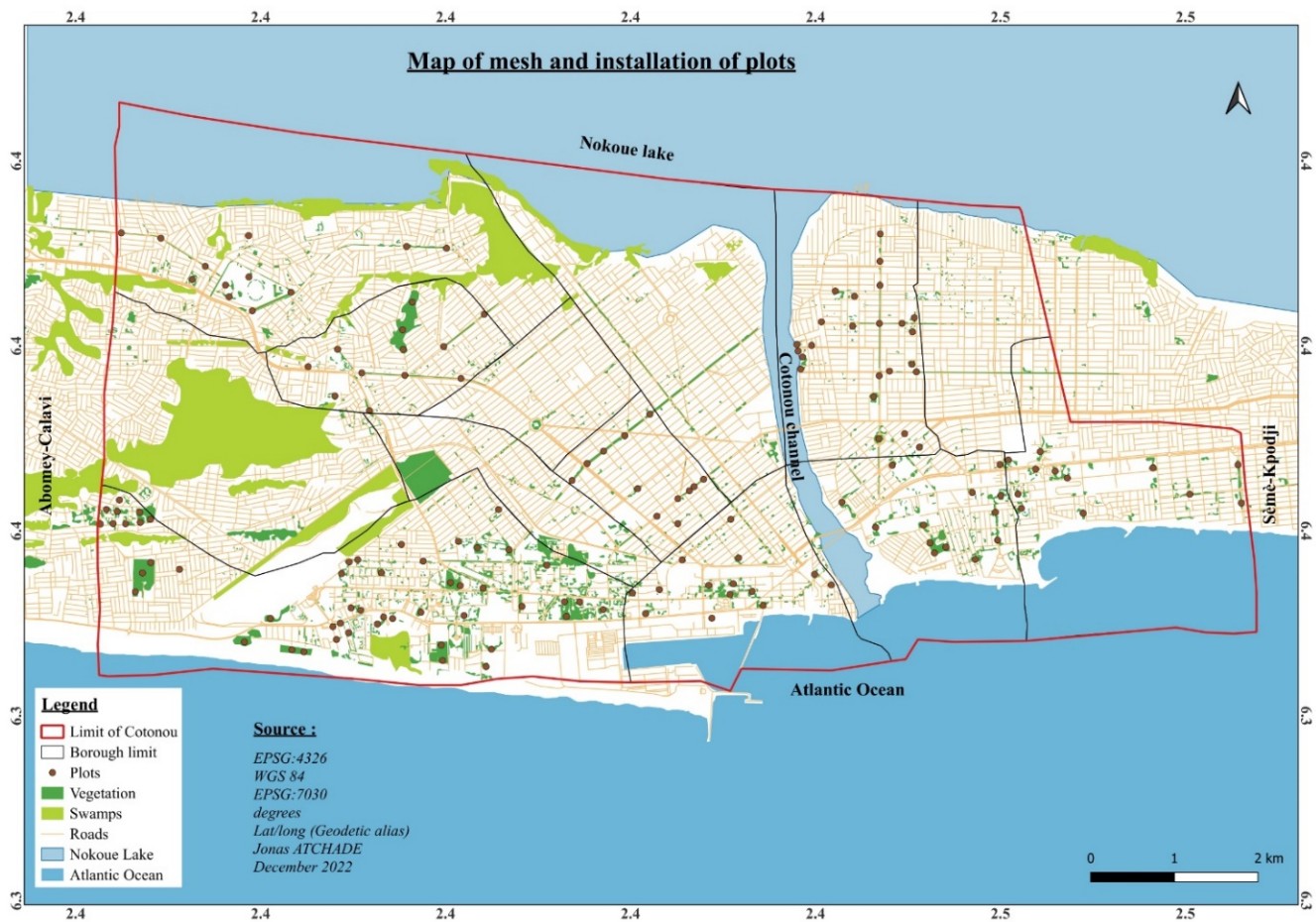

**Figure 2.** Map of the mesh and plot installation in the city of Cotonou.

### *2.3. Data Processing and Analysis*

#### 2.3.1. Taxonomic Diversity

Species whose scientific names were not known were identified using the flora of Benin [29]. Diversity parameters such as species richness (S), Shannon diversity index (H, Equation (1)), and Pielou diversity index (Eq, Equation (2)) were globally calculated for each type of land use. In addition, the number of families and genera were counted.

The relative frequencies of occurrence of species were calculated to highlight the most represented families and genera.

Specific richness (S): This is the number of species present in each area of the city of Cotonou. In this study, it is highlighted on each land use unit in the city.

Shannon diversity index (H): This index varies both in terms of the number of species present richness and in terms of the relative proportion of individuals of various species. It is expressed in bits. It is all the more important that a great number of species evenness participates in the recovery. That is to say, the higher its value, the greater the diversity. It is calculated according to the following formula:

$$H = -\sum pi \log 2pi \tag{1}$$

Pi (between 0 and 1): relative proportion of the number of individuals of a species i in the set of individuals of all species involved; Pi = ni/$\sum$n; with ni representing the number of individuals of species i and $\sum$n representing the set of individuals of all species.

Pielou's equitability: Pielou's equitability index (Eq) is the measure of the degree of diversity achieved by a plant grouping. It reflects the way individuals are distributed across plant groups. It varies between 0 and 1. Eq < 0.5 when there is a phenomenon of dominance (the individuals belong to a single species). Eq $\geq$ 0.5 when the distribution of individuals is homogeneous and results in an equitable distribution. Species are evenly distributed, and trees have the same dominance.

Equitability reflects the degree of diversity achieved by a stand and corresponds to the ratio between the effective diversity (H) and the theoretical maximum diversity (Hmax).

$$Eq = \frac{H}{Hmax} \text{with } Hmax = Log_2 S \tag{2}$$

In Equation (2), Hmax is the theoretical maximum Shannon value in each land use type. This value is at its maximum when species are identically abundant in each land use type and at its minimum when one species or a small group of species dominates in each land use type. The "vegan" package [30] was used to calculate the diversity indices in R. version 4.0.0 software.

2.3.2. Study of Species with High Ecological Importance in the City of Cotonou

The importance value index (IVI) of each land use type is calculated (Equation (3)). This index is used to determine the dominant species of each type of land use using (i) relative frequency, (ii) relative density, and (iii) relative dominance of the basal area (Table 1). The importance index value is between 0 and 300. Species with an IVI value greater than or equal to 10 were considered ecologically important species in the city of study. Frequency is calculated as the number of plots where a species is observed divided by the total number of plots surveyed. Relative frequency is calculated as frequency divided by the sum of the frequencies of all species multiplied by 100 (to obtain a percentage). Density is calculated as the total number of individuals of a species by a surface area. Relative density is calculated as density divided by the sum of the densities of all species multiplied by 100 (to obtain a percentage). Dominance is calculated as the total basal area of a species. Relative dominance is calculated by dividing the dominance by the sum of the dominances of all species multiplied by 100 to obtain a percentage (Table 1). For a species $\alpha$, the IVI is calculated according to Curtis et al. [31]:

$$IVI_{\alpha} = RD_{\alpha} + RF_{\alpha} + DoR_{\alpha} \tag{3}$$

In Equation (3), RD$\alpha$ is the relative density of species $\alpha$, RF$\alpha$ is the relative frequency of species $\alpha$, and DoR$\alpha$ is the relative dominance of species $\alpha$. With the value calculated from Table 1.

**Table 1.** Details of ecological parameters calculated to compare diversity in the different land use units of the city of Cotonou.

| Parameters | Formulars | Descriptions |
|---|---|---|
| Community density ($N$, tiges ha$^{-1}$) | $N = \frac{n}{s}$ | $n$: Total numbers of stems in the plot $s$: surface of the plot (ha$^{-1}$) |
| Basal surface ($G$, m$^2$ ha$^{-1}$) | $G = \frac{0.0001\ \pi}{4s} \sum_{i=1}^{n} d_i^2$ | $d_i$: diameter (cm) of the stem $i$ of the plot; $s$: area of plot in ha |
| Contribution to basal area ($Cs$, %) | $Cs = 100 \frac{Gpi}{G}$ | *Gpi:* basal area of the individuals of species $i$; $G$: basal area of the whole individuals of the plot |

2.3.3. Statistical Software

The data collected were first processed and formatted using a Microsoft 365 Excel spreadsheet. To carry out the statistical analyses, the biodiversity R package was used to calculate the various taxonomic diversity indices (specific richness, Shannon diversity index, and Pielou's equitability index). Then, statistical analyses were performed using the statistical software R (version 4.0.0, R Foundation for Statistical Computing, Vienna, Austria) [32]. The vegan package was used to calculate the various ecological diversity indices describing ecosystems. In this package, the various formulas described above were integrated as used by Peters et al. [33].

**3. Results**

*3.1. Floristic Diversity in the Land Use Units in the City of Cotonou*

A total of 1536 trees were visited and surveyed in 149 plots. Overall, 62 plant species of DBH ≥ 10 cm belonging to 55 genera and 27 families were counted. The total number of species inventoried and the floristic diversity in the city of Cotonou varies according to the types of land use in the city (Table 2). The analysis of Table 2 shows that in the city of Cotonou, residential areas recorded the highest number of plant species (S = 39) to 37 genera and 20 families, while the lowest number of species is recorded in commercial areas (S = 7) to 5 genera and 4 families. After residential areas (S = 39), administrative areas (S = 23) present the highest species richness followed by establishments (S = 14), roads (S = 11), green areas (S = 8), and commercial areas (S = 8). The values of the Shannon index and Piélou equitability vary from one land use unit to another and reflect the floristic diversity and the distribution of individuals by land use units.

**Table 2.** Parameters of floristic diversity of the inventoried species (S: specific richness, H: Shannon index, Eq: Piélou equitability, GE: number of genera, and FA: number of families) in the types of occupations on the grounds of the city of Cotonou.

| Type Occupation | Specific Richness (S) | Shannon Index (H) | Piélou Equitability (Eq) | Number of Genera (GE) | Number of Families (FA) |
|---|---|---|---|---|---|
| Administrative | 23 | 2.68 [a] | 0.86 [a] | 21 | 11 |
| Commercial | 7 | 1.59 [b] | 0.82 [a] | 5 | 4 |
| Green spaces | 8 | 1.58 [b] | 0.76 [a] | 7 | 4 |
| Establishment | 14 | 2.07 [b] | 0.79 [a] | 13 | 10 |
| Residence | 39 | 3.15 [a] | 0.86 [a] | 37 | 20 |
| Roads | 11 | 1.75 [b] | 0.73 [a] | 8 | 6 |
| **Total** | **62** | **2.86** | **0.69** | **55** | **27** |
| **Probability** | **-** | **<0.001** | **>0.5** | **-** | **-** |

Values with the same letter ([a] and [b]) are not significantly different at the 5% threshold. Values with different letters are significantly different at the 5% threshold (Scheffé's mean value structuring test).

Overall, Shannon's diversity index (H = 2.86 bits) and Piélou's equitability (Eq = 0.69) indicate a significant floristic diversity throughout the city. Observation of the results of the analysis of variance of the diversity parameters reveals significant differences (Table 2) between the city's land use units in terms of Shannon's diversity index ($p < 0.001$) in contrast to Piélou's equitability, which remains statistically similar ($p > 0.5$). From the comparative analysis, it appears that the highest Shannon diversity index value (H) is observed at the residential level (H = 3.15 bits), while the lowest value is recorded at the green space level (H = 1.58 bits) with some statistical significance ($p < 0.001$) at the 5% threshold. However, there is no significant difference between the Shannon diversity index values of the land use units such as residences (H = 3.15 bits) and administrations (H = 2.68 bits). The Shannon index is statistically similar for commercial areas (H = 1.59 bits), green spaces (H = 1.58 bits), roads (H = 1.75 bits), and establishment areas (H = 2.07 bits) and reflects low floristic diversity of these land use units compared to the administrative (H = 2.68 bits) and residential areas (H = 3.15 bits) (Table 2).

On the other hand, Table 2 shows that Pielou's equitability remains statistically similar ($p > 0.5$) between all land use units. This shows species evenness in all land use units in the city of Cotonou. Values with the same letter or combination of letters (a and b) are not significantly different at the 5% threshold (Scheffé's mean value structuring test).

Figure 3 illustrates the species richness by family (A) and by genus (B) in the city of Cotonou. From this figure, it appears that the Arecaceae have the most species with 14 species, followed, respectively, by the Fabaceae with 7 species, the Moraceae with 6 species, the Combretaceae with 4 species, and the Annonaceae with 3 species (Figure 3A). As for the genera, those most represented are, respectively, the Terminalia with 4 species, followed by the Artocarpus, Citrus, Ficus, Jatropha, and Phoenix. These last genera have two species each (Figure 3B).

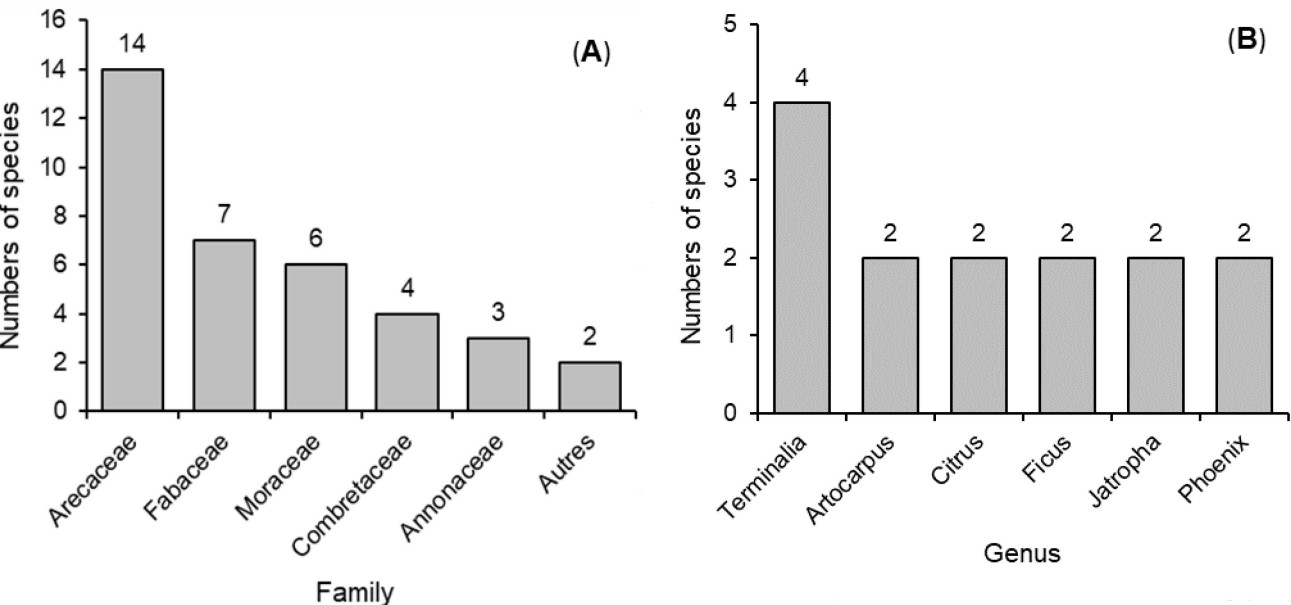

**Figure 3.** Illustration of species richness by family (**A**) and by genus (**B**) in the city of Cotonou. For family, others include the families of Anacardiaceae, Euphorbiaceae, Malvaceae, Meliaceae, Myrtaceae, Rutaceae, and Sapotaceae.

### 3.2. Top Five (5) Most Abundant Species in Land Use Units in the Cotonou City

The graphs in Figure 4 highlight the top five species by land use unit. The abscissa axis (species rank) expresses the top five species of the land use unit, while the ordinate axis expresses the abundance (in number of trees) of individuals of each species.

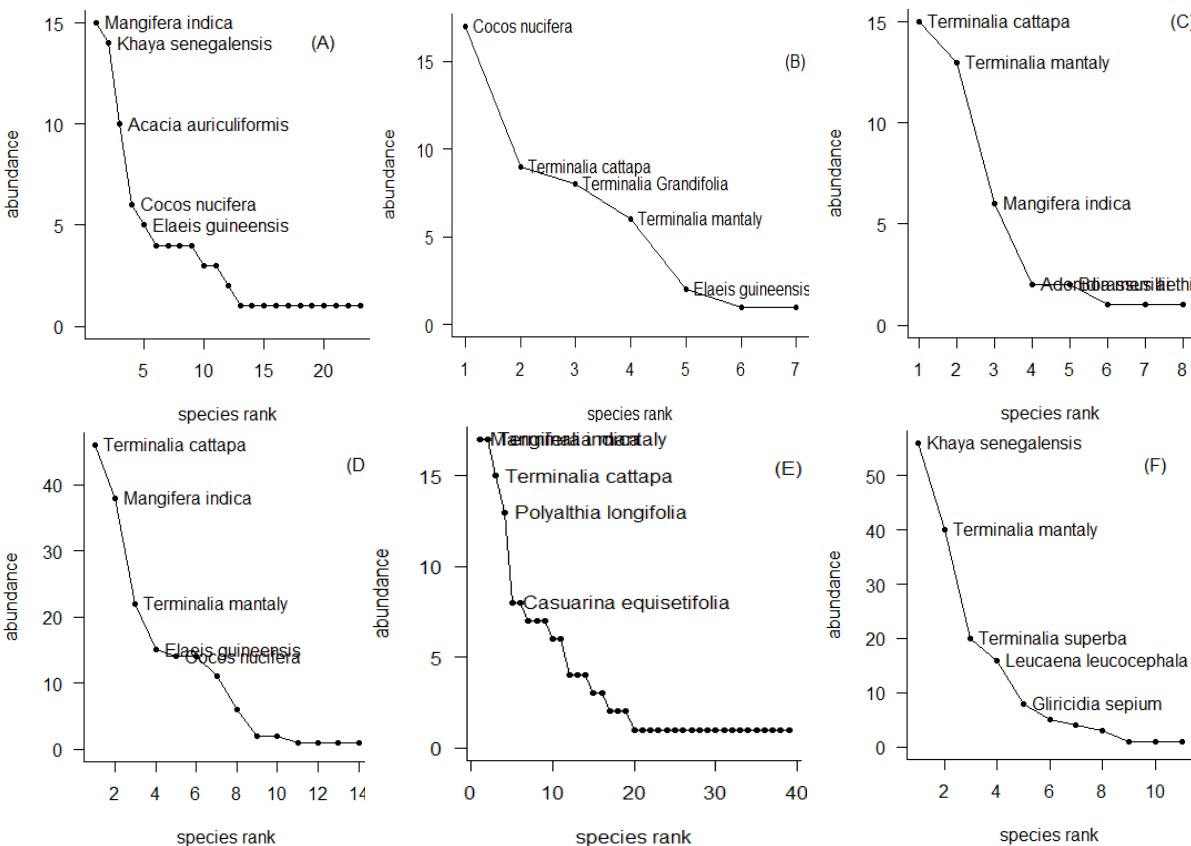

**Figure 4.** Rank-frequency curves of the top 5 most abundant species in land use types in the city of Cotonou. Administrative (**A**), commercial (**B**), green spaces (**C**), establishments (**D**), residences (**E**), and roads (**F**).

These graphs illustrate that the vegetation of all land use units in the city of Cotonou is strongly represented by a high frequency of exotic species.

The top five (5) most abundant species in the administrative zones (A) of the city are *Mangifera indica* (15), *Khaya senegalensis* (with respect to 14), *Acacia auriculiformis* (10), *Cocos nucifera* (6), and *Elaeis guineensis* (5). This compartment of the city represented by the public and private administration is taken over by species that are mostly edible and exotic. *Khaya senegalensis* (with 14 individuals) is the only resistant native species with a strong representation. At the level of the commercial centers (B) of the study city, we note the strong presence of species such as *Cocos nucifera* (15 individuals), *Terminalia catappa* (9 individuals), *Terminalia grandifolia* (8 individuals), *Terminalia mantaly* (6 individuals), and *Elaeis guineensis* (3 individuals). The abundant presence of these species justifies the ecosystem services provided in the city. The green spaces (C), which are represented by the woodlands and urban forests, botanical gardens, interstices and cemeteries, and the peripheral zone, are mainly full of *Terminalia Catappa* (15), *Terminalia mantally* (13), *Mangifera indica* (6), *Adonidia merrillii* (2), and *Borassus aethiopum*. The establishments (D), which are mainly training and apprenticeship centers, are peaked by the majority of species such as *Terminalia catappa* (48), *Mangifera indica* (38), *Terminalia mantaly* (22), *Elaeis guineensis* (15), and *Cocos nucifera* (13). The first three most abundant species define the potential regulating services offered in this land use unit of the city. The residential compartment (E) is mainly and respectively full of species such as *Terminalia mantaly* (19), *Mangifera indica* (19), *Terminalia catappa* (15), *Polyalthia longifolia* (13), and *Casuarina equisetifolia* (8).

At the level of the road (F), the abundance of species reveals that the species *Khaya senegalensis* (59) is the most abundant followed by *Terminalia mantaly* (40), *Terminalia superba* (20), *Leucaena leucocephala* (18), and *Gliricidia sepium* (8).

### 3.3. Ecological Importance of Species in the City of Cotonou

The importance value index (IVI) characterizes the importance within a stand of a species relative to the total of all species present in the vegetation under consideration. Table 3 shows the ecologically important species (IVI ≥ 10%) by land use unit. The analysis of Table 3 shows the five (5) species with the highest importance values (IVI) in the land use units of the city of Cotonou. From this analysis, it appears that *Mangifera indica* (IVI = 64.06%) is the most ecologically important plant species in the administrative zones, while *Terminalia catappa* (121.47%) occupies this position in the commercial zones of the city. Regarding the green areas and establishments, *Terminalia catappa* remains the most ecologically important plant species with 113.15% and 86.16% as the values of the ecological importance index respectively. In relation to the land use unit's residential areas and roads, *Mangifera indica* (52.06%) and *Khaya senegalensis* (151.16%) are respectively the most ecologically important plant species. Not only did *Terminalia catappa*, an exotic species, rank among the top five species of high ecological importance in all land use units except for the administrative zones but it should also be noted that the species occupies first place in three of the six land use units studied in the city of Cotonou. On the other hand, *Khaya senegalensis*, even though the native species is present in the top five ecologically important species in all land use units except for the commercial areas, only has the highest IVI value in one land use unit (road). As for *Mangifera indica*, the exotic species occupies the first place in two land use units and is present in the top five ecologically important species in five of the six land use units. Based on the Table 4, all of this shows that exotic plant species have taken over and predominated in the city's land use units and that this guides the ecological development policies of Benin's cities.

**Table 3.** List of the five species with the highest ecological importance value index in the land use types of the city of Cotonou. RDα is the relative density of each (α) species, RFα is the relative frequency of species α, and DoRα is the relative dominance of species α.

| Land Use Units | Species | RFα (%) | RDα (%) | DoRα (%) | IVI (%) |
|---|---|---|---|---|---|
| Administrative | *Mangifera indica* | 10.34 | 20.55 | 33.17 | 64.06 |
| | *Khaya senegalensis* | 10.34 | 19.18 | 30.17 | 59.69 |
| | *Acacia auriculiformis* | 10.34 | 13.70 | 4.33 | 28.37 |
| | *Terminalia mantaly* | 3.45 | 5.48 | 10.29 | 19.22 |
| | *Elaeis guineensis* | 6.90 | 6.85 | 3.98 | 17.73 |
| Commercial | *Terminalia cattapa* | 25.00 | 30.00 | 66.47 | 121.47 |
| | *Cocos nucifera* | 37.50 | 56.67 | 18.82 | 112.99 |
| | *Mangifera indica* | 12.50 | 3.33 | 13.03 | 28.87 |
| | *Elaeis guineensis* | 12.50 | 6.67 | 0.86 | 20.03 |
| | *Acacia auriculiformis* | 12.50 | 3.33 | 0.81 | 16.65 |
| Green spaces | *Terminalia cattapa* | 30.00 | 38.46 | 44.69 | 113.15 |
| | *Terminalia mantaly* | 20.00 | 33.33 | 37.17 | 90.50 |
| | *Mangifera indica* | 10.00 | 15.38 | 16.37 | 41.75 |
| | *Borassus aethiopum* | 10.00 | 5.13 | 0.20 | 15.33 |
| | *Khaya senegalensis* | 10,00 | 2.56 | 1.48 | 14.04 |
| Establishment | *Terminalia cattapa* | 23.40 | 27.61 | 35.15 | 86.16 |
| | *Mangifera indica* | 17.02 | 23.31 | 30.13 | 70.47 |
| | *Terminalia mantaly* | 14.89 | 13.50 | 15.82 | 44.21 |
| | *Khaya senegalensis* | 8,51 | 8.59 | 11.08 | 28.18 |
| | *Elaeis guineensis* | 12,77 | 9.20 | 4.83 | 26.80 |
| Residence | *Mangifera indica* | 12.00 | 14.17 | 25.90 | 52.06 |
| | *Terminalia mantaly* | 6.00 | 14.17 | 26.02 | 46.18 |
| | *Terminalia cattapa* | 8.00 | 12.50 | 21.19 | 41.69 |
| | *Khaya senegalensis* | 6.00 | 6.67 | 12.89 | 25.56 |
| | *Polyalthia longifolia* | 6.00 | 10.83 | 3.63 | 20.46 |

**Table 3.** *Cont.*

| Land Use Units | Species | RFα (%) | RDα (%) | DoRα (%) | IVI (%) |
|---|---|---|---|---|---|
| | *Khaya senegalensis* | 36.36 | 54.37 | 60.43 | 151.16 |
| | *Terminalia mantaly* | 31.82 | 28.16 | 27.40 | 87.37 |
| Roads | *Terminalia cattapa* | 13.64 | 4.85 | 568 | 24.17 |
| | *Leucaena leucocephala* | 4.55 | 7.77 | 5.80 | 18.11 |
| | *Acacia auriculiformis* | 4.55 | 2.91 | 4.61 | 12.07 |

**Table 4.** List of species (including genus and family) inventoried in urban Cotonou.

| Species | Genus | Family |
|---|---|---|
| *Acacia auriculiformis* | Acacia | Fabaceae |
| *Adonidia merrillii* | Adonidia | Arecaceae |
| *Anacardium occidentale* | Anacardium | Anacardiaceae |
| *Annona muricata* | Annona | Annonaceae |
| *Archontophoenix cunninghamiana* | Archontophoenix | Arecaceae |
| *Areca catechu* | Areca | Arecaceae |
| *Arenga pinnata* | Arenga | Arecaceae |
| *Artocarpus altilis* | Artocarpus | Moraceae |
| *Artocarpus heterophyllus* | Artocarpus | Moraceae |
| *Azadirachta indica* | Azadirachta | Meliaceae |
| *Blighia sapida* | Blighia | Sapindaceae |
| *Borassus aethiopum* | Borassus | Arecaceae |
| *Calotropis procera* | Calotropis | Asclepiadaceae |
| *Carica papaya* | Carica | Caricaceae |
| *Caryota urens* | Caryota | Arecaceae |
| *Casuarina equisetifolia* | Casuarina | Casuarinaceae |
| *Ceiba pentandra* | Ceiba | Malvaceae |
| *Chrysophyllum albidum* | Chrysophyllum | Sapotaceae |
| *Citrus sinensis* | Citrus | Rutaceae |
| *Citru* sp. | Citrus | Rutaceae |
| *Cocos nucifera* | Cocos | Arecaceae |
| *Delonix regia* | Delonix | Fabaceae |
| *Dypsis lutescens* | Dypsis | Arecaceae |
| *Elaeis guineensis* | Elaeis | Arecaceae |
| *Eucalyptus camaldulensis* | Eucalyptus | Myrtaceae |
| *Eucalyptus torrelliana* | Eucalyptus | Myrtaceae |
| *Ficus benjamina* | Ficus | Moraceae |
| *Ficus* spp. | Ficus | Moraceae |
| *Gliricidia sepium* | Gliricidia | Fabaceae |
| *Gmelina arborea* | Gmelina | Lamiaceae |
| *Hyphaene thebaica* | Hyphaene | Arecaceae |
| *Irvingia gabonensis* | Irvingia | Irvingiaceae |
| *Jatropha curcas* | Jatropha | Euphorbiaceae |
| *Jatropha integerrima* | Jatropha | Euphorbiaceae |

**Table 4.** *Cont.*

| Species | Genus | Family |
|---|---|---|
| *Khaya senegalensis* | Khaya | Meliaceae |
| *Leucaena leucocephala* | Leucaena | Fabaceae |
| *Mangifera indica* | Mangifera | Anacardiaceae |
| *Moringa oleifera* | Moringa | Moringaceae |
| *Musa* spp. | Musa | Musaceae |
| *Newbouldia laevis* | Newbouldia | Bignoniaceae |
| *Persea americana* | Persea | Lauraceae |
| *Phoenix dactylifera* | Phoenix | Arecaceae |
| *Phoenix reclinata* | Phoenix | Arecaceae |
| *Polyalthia longifolia* | Polyalthia | Annonaceae |
| *Pouteria alnifolia* | Pouteria | Sapotaceae |
| *Prosopis africana* | Prosopis | Leguminosae-Mim |
| *Psidium guajava* | Psidium | Myrtaceae |
| *Pterocarpus erinaceus* | Pterocarpus | Fabaceae |
| *Raphia hookeri* | Raphia | Arecaceae |
| *Ravenala madagascariensis* | Ravenala | Strelitziaceae |
| *Roystonea regia* | Roystonea | Arecaceae |
| *Salix babylonica* | Salix | Salicaceae |
| *Sebestenia cordia* | Sebestenia | Euphorbiaceae |
| *Senna siamea* | Senna | Fabaceae |
| *Spondias mombin* | Spondias | Anacardiaceae |
| *Tectona grandis* | Tectona | Lamiaceae |
| *Terminalia cattapa* | Terminalia | Combretaceae |
| *Terminalia grandifolia* | Terminalia | Combretaceae |
| *Terminalia mantaly* | Terminalia | Combretaceae |
| *Terminalia superba* | Terminalia | Combretaceae |
| *Thevetia peruviana* | Thevetia | Apocynaceae |
| *Treculia africana* | Treculia | Moraceae |

## 4. Discussion

### 4.1. Floristic Composition and Specific Diversity of the Urban Vegetation

Over all the plots, 62 species divided into 55 genera and 27 families were inventoried. Thus, these results are in line with those of other researchers in West African cities. For instance, the number of species recorded in Cotonou is close to that obtained by Amontcha [27] in the city of Porto-Novo (73 species). The results of [9] revealed 67 species divided into 54 genera and 28 families in the city of Atakpamé (Togo). Our results reinforce those of these author's work on trees' diversity in urban areas. Contrary to Amontcha [27], Sehoun et al. [19] found 22 tree species in their scientific work on the city of Cotonou. These authors only considered public and private green spaces, while our investigation considered all the land use units of the city. On the other hand, in research titled "Structure, diversity and carbon stocks of trees in the city of Kumasi in Ghana", Nero et al. [34] obtained 176 species and 42 families in the different types of urban trees in the city of Kumasi. In the city of Lomé in Togo, higher values than ours were recorded by Polorigni et al. [16] in the city of Lomé, with 93 species and 47 families. These results show a high diversity

of tree species in the cities of Kumasi and Lomé in contrast to the city of Cotonou, which records more, but less diversified, trees. The low diversity values of urban vegetation in the city of Cotonou could be explained not only by the climatic and soil conditions that are not very favorable to the survival of several plant species but also by the city's ecological development policies as mentioned by the work of [34]. This could also be the result of different urban forest management (i.e., planting, removals, etc.) across the various boroughs of the city, varying levels of funding/staffing for urban forestry programs, potential natural disasters that may have impacted an area, local policies/ordinances, development codes, tree protection codes, mitigation efforts, landscape designs, etc. The author links this to the relatively favorable climatic conditions in certain urban environments. In fact, some authors believe that urban forestry planning must take climate variability and change into account in order to provide an ideal living environment in a context where the uncertainty surrounding climate change and its effects does not make life easy for city dwellers. That is why Li et al. [14] revealed that species diversity among different cities is affected by climate and topography as well as human factors. This could be explained by the fact that in the city of Cotonou, horticultural species are not most represented, as shown by Polorigni et al. [16] and Radji et al. [23], in the species plots.

*4.2. Specific Diversity of Vegetation, Abundance of Exotic Species, and Ecological Importance in the Land Use Units*

The analysis of variance of the diversity parameters reveals significant differences (Table 2) between the land use units of the city in terms of the Shannon diversity index ($p < 0.001$) in contrast to the Piélou equitability, which remains statistically similar ($p > 0.5$). In the city of Cotonou, residential (H = 3.15 bits), administrative (H = 2.68 bits), and settlement areas are the occupancy units that recorded the highest values of the Shannon index and testify to the plant diversity of these occupancy units. The work of Orou [21] in the city of Malanville revealed that residential areas hold the highest Shannon index value. The results of Nero et al. [34] on the city of Kumasi in Ghana show that administrative areas are among the most floristically diverse occupancy units. In Togo, Folega et al. [9] prioritized settlements as places among the most floristically diverse places in terms of urban flora. The species richness and Shannon index, varying from one occupancy unit to another, express the floristic diversity of the city's occupancy units. This was highlighted by Xin et al. [35] in their study on the analysis of the effects of ecological conservation redline policies in the Pearl River Delta area, China. They found that Shannon's diversity index on the inside of the city is lower than in the outside area, which indicates a high degree and rapid landscape fragmentation in the outside area The diversity and richness of flora in residential areas could be justified by the arrangements that city dwellers make to introduce specific species into their homes based on preferences for the ecosystem services like regulation services (shake, carbon storage, etc.), provisioning services (food, medicinal use, etc.) and cultural services these species provide. Urban biodiversity is often attributed to the inherent and preferential location of cities in biodiversity hotspots, socio-ecological factors, and human actions through species introduction and landscape heterogeneity [34]. Residential areas and administrative complexes reflect the ecological values, variety of interests, and socioeconomic status of the owners and users of these spaces. The high species diversity in residential areas is a manifestation of their multifunctional and structural complexity [36] underpinned by a variety of socioecological constraints [37]. In contrast to home gardens, which are predominantly private, vegetation in institutional compounds is primarily maintained by government administrative authorities for its qualities of shade and ornament, boundary demarcation, windbreak, and environmental protection, as well as sometimes for food reasons.

The study shows that exotic species such as *Terminalia Catappa*, *Terminalia mantaly*, and *Mangifera indica* are very abundant in four out of six land use units in the city of Cotonou and outnumber the other species in terms of abundance. *Khaya senegalensis*, a native species, is present and abundant in only two of the six land use units visited in

Cotonou. These results corroborate those of Folega et al. [9], who concluded that the vegetation in the city of Atakpamé is represented by a high frequency of exotic species. The abundance of widely distributed species (pantropical and introduced species) indicates that the study area belongs to the highly anthropized domain disturbed by phenomena catalyzed by the urbanization gradient [19,38]. Only one dominant endogenous species (*Khaya senegalensis*) of high ecological importance tops a single occupancy unit of the six cities. This result agrees with Polorigni et al. [16] and Osseni et al. [39], according to whom exotic species predominate over urban vegetation. According to [40], populations settle, fragment, and degrade or destroy natural areas, thus disappearing or losing local or indigenous biodiversity and introducing exotic species for their comfort. The results from Teka et al. [20] showed that street alignment trees were poorly diversified and dominated by *Khaya senegalensis* in Cotonou. This species appeared to be highly pruned and threatened because of its numerous medicinal virtues. Moreover, it was found that air temperature and relative humidity were influenced by this species. The same authors found that the coolness effect of urban green was evidenced by the decrease in temperature under alignment trees compared with that recorded on roadsides free of trees. The city's ecological development policies have catalyzed the invasion of all land use units by exotic plant species. These exotic tree species are better adapted to the extreme weather conditions in Africa, especially in cities. Then, many of them are valued by urban authorities. However, it is worth noting that these species, in some cases, are outcompeted in relation to native species, which are slowly disappearing from the urban landscape [36]. The indigenous species that are resilient in the face of climate change or harsh conditions in the city and that are found in the top five plant species of high ecological importance are hosted by a single land use unit (the roads). However, the presence of certain exotic plant species in certain land use units supports highly desired specific ecosystem services offered by floral diversity in the city of Cotonou. According to Osséni et al. [39], the preponderance of these species has been favored by their promotion as species adapted to the urban environment and biophysical conditions including climate variability. They have been used during reforestation and regreening campaigns for the past three decades. Species in the occupation units would be explained by the fact that most of the species are better adapted to the climatic and edaphic conditions of the city [33,41]. In addition, these species are evergreen trees that can provide certain ecosystem services such as shade in all seasons [21,42]. This is, moreover, what could explain the high proportion of these species in urban areas. Others are found there for their beautiful flowers. The choice of planting indigenous tree species in the city would facilitate the conservation of local plant diversity in urban ecosystems. The *Khaya senegalensis* species, for instance, is very resilient to climatic downpours and produces many ecosystem services, mainly supplying services such as the medicinal virtues frequently sought by city dwellers to combat malaria. The IVI lies in its potential to guide and redirect policy decisions in ecological land use planning.

## 5. Conclusions

This study provided a range of data on the tree species present in the urban and peri-urban flora of the city of Cotonou. The diversity in species ensures that a wide range of ecosystem services are provided across the city. This study reveals that residential areas, administrations, and establishments are the land use units that have recorded the highest values of the Shannon index and are the most diverse in terms of urban flora. This tree species is characterized by a strong preponderance of exotic species such as *Terminalia Catappa, Terminalia mantaly,* and *Mangifera indica*. An indigenous species strongly resilient to the extreme weather and anthropogenic threats resulting from climate change and urbanization of the urban soils is *Khaya senegalensis*. This species is of high ecological importance with high IVI values across all land uses and offers diversified ecosystem services to the city's residents. All this significant ecological potential should be developed to make a strong contribution to nature-based solutions in urban areas in the face of the adverse effects of climate change. The information generated from the study can be used to

formulate guidelines, policies, or strategies so that the selective development of this urban vegetation can improve the resilience of urban life to climate risks through the provision of urban ecosystem services, possible foundations of ecological infrastructures, and urban nature-based solutions. In view of this opportunity, it would be wise for all stakeholders in urban management to join forces to promote ecologically sustainable infrastructure to facilitate the achievement of the Sustainable Development Goals (SDGs), in this case SDG 11, in African cities. This multi-stakeholder approach could enable the country to honor its international commitments to the climate and the planet. We cannot conclude this study without making a number of recommendations to public and urban authorities in Africa. These recommendations concern the prospects for future scientific investigations to not only value urban tree services but also provide human services to urban trees to protect them from climate change impacts. Future priorities include funding urban ecology research to understand the resilience of urban tree species in the context of climate change, and the factors favoring the coexistence of indigenous and exotic species, in order to take advantage of the ecosystem services likely to offer climate resilience to city dwellers.

**Author Contributions:** Conceptualization, A.J.A.; Methodology, A.J.A. and M.K.; Software, A.J.A.; Validation, K.A.; Formal analysis, A.J.A. and M.K.; Investigation, A.J.A.; Resources, M.K. and K.A.; Data curation, A.J.A. and H.Y.; Writing—original draft, A.J.A.; Writing—review & editing, M.K.; H.Y. and F.F.; Visualization, M.K., F.F. and K.W.; Supervision, M.D., K.W. and K.A. All authors have read and agreed to the published version of the manuscript.

**Funding:** This research was funded by the Regional Centre of Excellence on Sustainable Cities in Africa (CERViDA_DOUNEDON), the Association of African Universities (AUA), and the World Bank Group. This paper is a part of PhD data collection results whose the amount is 4943.49 USD.

**Institutional Review Board Statement:** Not applicable.

**Informed Consent Statement:** Not applicable.

**Data Availability Statement:** Data will be made available on request.

**Acknowledgments:** We are grateful to the Regional Centre of Excellence on Sustainable Cities in Africa (CERViDA_DOUNEDON), the Association of African Universities (AUA), and the World Bank for providing the necessary funding that facilitated our research work leading to these results. We would also like to express our gratitude to Cyprien Aholou, Kossiwa Zinsou, Epse Klassou, and Yanik Akin for all their support in this scientific journey.

**Conflicts of Interest:** The authors declare no conflict of interest.

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
