# Peer review of "Trees Diversity and Species with High Ecological Importance for a Resilient Urban Area: Evidence from Cotonou City (West Africa)"

_climate, doi:10.3390/cli11090182_

Round 1
Reviewer 1 Report (New Reviewer)
I consider the article an interesting topic although that the publication needs some improvements both to structure and content.
Here are some observations and suggestions to authors to take into consideration for a good revision of the manuscript:
In all the manuscript, refer to the name of the author before signaling the number of the reference. Example: According to Atchade et al. (1) instead of according to (1).
“In the city of Cotonou” is repeated many times throughout the manuscript, it could be removed from the text at the exception of material and methods, figures and tables.
Abbreviation: There is a need to add abbreviation section to avoid repetitions
7-10 There is no need for the words “Author from the”
116-118 the repetition should be avoided “we randomly selected 8 of the 13”
133- …. “respectively” were set up.
137: in each plot, ..
139-140, 211: diameter at breast height (DBH), to use the same abbreviation usually..
153 Shannon diversity index (H): This index varies…..
159: to revise the formula!
164,166: to avoid repetition, to use just “Eq”
168,169: to formulate to just one sentence!
172 with to replace Avec
186 Density is calculated as the total number of individuals of a species.by a surface area
Results: THE SECTIONS 2-2 and 2-3 could me merged into one section
213, 343: table 2 (remove II)
214: belonging to 37 genera
216 belonging to 5 genera
229: no need of “statistically” word
241: there is no letter “c” in the table. To remove from the description of the table.
254: …. and Sapotaceae.
290: five (5)
294: “About of”: replace by “with respect to”
Discussion
318. Our results reinforce those of these author’s work……
320. This discrepancy between the number of plant species in their work….
The number of species in your study is not supposed to be the same as other studies (descriptive parameters): I do not see the comparison here useful when discussing results, you can just mention the diversity in other studies and discuss and add your comments….
392;393 However, it's worth noting that these species, in some cases are repressive in relation to native species, which are slowly disappearing from the urban landscape [17].
The direct effect of exogenous species on indigenous species needs more deep explanations: It is not sufficient to affirm that the diversity or frequency of native species is low because of the introduction of exotic plant species….
317: in the of the city of Porto-Novo in the city of ..
340: and
369: wind break, to cut wind: it’s sufficient just one expression
408: There is no need to repeat, just mention the “Ecological Importance Value Index (IVI)” and then use the abbreviation
Figures: The quality of graphs could be improved, horizontal lines in the figure 3 could be eliminated, Genus in cursive, …
Conclusions:
412 It allowed us to identify 62 tree species divided into 55 genera 412 and 27 families. NOT INTERESTING SENTENCE IN CONCLUSIONS
411: specify which ability
References
This section should be REWRRITEN according to the author instructions available on the webpage of Climate journal
https://www.mdpi.com/journal/climate/instructions
Author Response
Dear Reviewer
Thank you for taking the time to provide quality comments on our manuscript despite your busy schedule.
Taking into account your comments, please find the answers point by point in the lines below.
Point by pont responses
Point 1
1-In all the manuscript, refer to the name of the author before signaling the number of the reference. Example: According to Atchade et al. (1) instead of according to (1).
“In the city of Cotonou” is repeated many times throughout the manuscript, it could be removed from the text at the exception of material and methods, figures and tables.
Abbreviation: There is a need to add abbreviation section to avoid repetitions
Response 1
This suggestion has been addressed as recommended throughout all the manuscript (you can take a look and confirm).
Point 2
7-10 There is no need for the words “Author from the”
Response 2
This has been addressed in the revised version.
Point 3
116-118 the repetition should be avoided “we randomly selected 8 of the 13”
Response 3
This is addressed in the revised version of the manuscript.
Point 4
133- …. “respectively” were set up.
Responses 3
The word RESPECTIVELY has been integrated in the sentence as suggested.
Point 5
137: in each plot, ..
Response 5
It is done as suggested.
Point 6
139-140, 211: diameter at breast height (DBH), to use the same abbreviation usually..
Response 6
Corrected as recommended.
Point 7
153 Shannon diversity index (H): This index varies…..
Response 7
Corrected in the revised version of the manuscript.
Point 8
159: to revise the formula!
Response 8
The formula is revised as suggested, with the emphasis on Pi=ni/n; with ni: number of individuals of species i and ∑ n: set of individuals of all species.
Point 9
164,166: to avoid repetition, to use just “Eq”
Response 9
This is corrected in the revised version.
Point 10
168,169: to formulate to just one sentence!
Response 10
This is done as recommended.
Point 11
172 with to replace Avec
Response 11
Correted.
Point 12
186 Density is calculated as the total number of individuals of a species.by a surface area
Response 11
It is completed by a surface area as recommended.
Point 12
Results: THE SECTIONS 2-2 and 2-3 could me merged into one section
Response 12
The two sections are mergerd as recommended by the reviewer.
Point 13
213, 343: table 2 (remove II)
Response 13
It is donne as suggested.
Point 14
213, 343: table 2 (remove II)
Response 14
This is removed from the revised version.
Point 15
214: belonging to 37 genera
Response 15
From was replaced by belonging to.
Point 16
216 belonging to 5 genera
Response 16
From was replaced by belonging to.
Point 17
229: no need of “statistically” word
Response 17
Statistically is removed from the updated version.
Point 18
241: there is no letter “c” in the table. To remove from the description of the table.
Response 18
It is done as recommended.
Point 19
254: …. and Sapotaceae.
Response 19
It is done asrecommended.
Point 20
290: five (5)
Response 20
It is added as suggested
Point 21
294: “About of”: replace by “with respect to”
Response 21
It is replaced as recommended.
About the Discussion
Point 22
- Our results reinforce those of these author’s work……
Response 22
Our results reinforce those of these author’s work… on trees diversity in urban area (this is completed in the revised version of the manuscript.
Point 23
- This discrepancy between the number of plant species in their work….
Response 23
This is reformulated as: This discrepancy between the number of plant species in their work and ours could be explained by the fact that these authors only considered public and private green spaces, while our investigation considered all the land use units of the city.
Point 24
320. The number of species in your study is not supposed to be the same as other studies (descriptive parameters): I do not see the comparison here useful when discussing results, you can just mention the diversity in other studies and discuss and add your comments….
Response 24
This is right, but we were trying to point out the species number in our study area with others. And as you recommended, this is canceled to make the sentence clear.
Point 25
392;393 However, it's worth noting that these species, in some cases are repressive in relation to native species, which are slowly disappearing from the urban landscape [17].
Response 25
This is corrected and reformulated in the last version of the manuscript.
Point 26
The direct effect of exogenous species on indigenous species needs more deep explanations: It is not sufficient to affirm that the diversity or frequency of native species is low because of the introduction of exotic plant species….
Response 26
Indeed, this is not normally the only raison, but we mentioned this as one of the several raisons, as this demontrated by others study. Meanwhile, we have reformulated this sentence.
Point 27
317: in the of the city of Porto-Novo in the city of ..
Response 27
Corrected in the new draft.
Point 28
340: and
Response 28
Corrected
Point 29
369: wind break, to cut wind: it’s sufficient just one expression
Response 29
Corrected in the last version.
Point 30
408: There is no need to repeat, just mention the “Ecological Importance Value Index (IVI)” and then use the abbreviation
Response 30
Corrected.
Point 31
Figures: The quality of graphs could be improved, horizontal lines in the figure 3 could be eliminated, Genus in cursive, …
Response 31
Figure 3 has been reworked with the suggestions in mind, and the quality is once again very good.
About the conclusion
Point 32
412 It allowed us to identify 62 tree species divided into 55 genera 412 and 27 families. NOT INTERESTING SENTENCE IN CONCLUSIONS
Response 32
The sentence si canceled and some parts of the conclusion have been improved suggested.
Point 33
411: specify which ability
Response 33
This is donne.
About the References
Point 34
This section should be REWRRITEN according to the author instructions available on the webpage of Climate journal.
Response 34
All the section is reformulated as recommended by MDPI reference guidanvce indicates. You can see that in the new version of the manuscript.
We can see that your comments and suggestions have much more to do with the form of the manuscript. We have addressed them point by point as recommended. We hope this meets your expectations. We remain available to provide further clarification if required.
Regards.

Reviewer 2 Report (New Reviewer)
Overall, the manuscript is well-written, and the study team did a nice job with explaining the methods and analyzing the data. Beyond the minor suggestions listed below, the manuscript would benefit from the acknowledgement of the limitations of the study in the discussion or the conclusion. Likewise, a more detailed explanation of the impacts of the findings as it relates to urban forest management in Cotonou and West Africa as a whole.
[Line 16] – Delete extra period after “Cotonou..”
[Lines 32-33] – Keywords: Urban trees diversity; ecosystem services; ecological importance; species abundance; urban ecological planning; climate adaptation; Conservation planning; Cotonou
Drop the following as they are already in the title – urban tree diversity, ecological importance, and Cotonou.
Make lowercase – Conservation.
[Line 40] – Delete the word “living”.
[Line 42] – Remove the “-“ between “ment-both” and “components-will”.
[Line 54] – Change “environment” to “environmental”
[Line 56] – It’s unclear what is meant by “natural balance, species restoration,”. Consider rewording both.
[Line 71] – Delete “of”.
[Line 137-138] – Change sentence from “This concerns the scientific names, families and genera of the species.on the one hand.” To “On the one hand, this concerns the scientific names, families and genera of the species.”
[Lines 140 and 211] It says “DHP”, did you mean DBH? If so change, if not, define DHP.
[Line 151] – Is it “Specific richness (S)” or “Species richness (S)”?
[Lines 153-154] – Put “Shannon diversity index (H)” on the same line as “The Shannon diversity index varies both in terms of the number of species present” as you did with “Species richness (S)” on line 151.
[Line 163-164] – Same comment as above, move “Pielou's equitability (Eq)” to the same line as it’s description.
[Line 178] – Sentence begins with a spiral symbol, delete unless this was intentional.
[Line 191] – Delete “()” from “(to obtain a percentage)”.
[Line 199] – Sentence begins with a spiral symbol, delete unless this was intentional.
[Line 213] – Change “Table II” to “Table 2”.
[Line 216] – Change “(S=39);” to “(S=39),”.
[Line 251] – The text on Figure 3 is blurry, replace with a higher resolution image.
[Line 253-254] – Add comas and an “and” to the sentence “……the families of Anacardiaceae Euphorbiaceae Malvaceae Meliaceae Myrtaceae Rutaceae Sapotaceae.” It should then read “…..the families of Anacardiaceae, Euphorbiaceae, Malvaceae, Meliaceae, Myrtaceae, Rutaceae, and Sapotaceae.”
[Line 283] – Graphs C, D, and E in Figure 4 all have species names that are on top of each other, making it difficult to read. Try to update the three graphs that the species are no longer depicted with the words on top of one another.
[Line 294] – Change sentence from “About of green areas and etablishments,….” To “Regarding the green areas and establishments,…..”.
[Line 319-331] – Could this not also be the result of different urban forest management (i.e., planting, removals, etc.) across the various cities, varying levels of funding/staffing for urban forestry programs, potential natural disasters that may have impacted an area, local polies/ordinances, development codes, tree protection codes, mitigation efforts, landscape designs, etc. Thus, climatic conditions are just one of many variables that could affect species composition and diversity. It would be of value to make mention of this.
[Line 373] – Change “land-use” to “land use”.
[Line 374] – delete the work “of”.
[Line 399] – Add “and” between “urban environment” and “biophysical conditions”.
[Line 434] – Change “These concerns:-funding urban ecology research….” To “These include funding urban ecology research……..”
[Line 435] – Remove extra spaces between “tree species in the context”.
[Line 436] – Change “-the factors……” to “the factors…..”.
[Line 573] – Delete “46.”.
See my general comments/suggestions to the authors.
Author Response
Dear Reviewer
Thank you for taking the time to provide quality comments on our manuscript despite your busy schedule.
Taking into account your comments, please find the answers point by point in the lines below.
Point by pont responses
Point 1: [Line 16] – Delete extra period after “Cotonou..”
Response 1: done
Point 2: [Lines 32-33] – Keywords: Urban trees diversity; ecosystem services; ecological importance; species abundance; urban ecological planning; climate adaptation; Conservation planning; Cotonou
Drop the following as they are already in the title – urban tree diversity, ecological importance, and Cotonou.
Make lowercase – Conservation.
Response 2: Done
Point 3: [Line 40] – Delete the word “living”.
Response 3: done
Point 4: [Line 42] – Remove the “-“ between “ment-both” and “components-will”.
Response 4: Done
Point 5: [Line 54] – Change “environment” to “environmental”
Response 5: done
Point 6: [Line 56] – It’s unclear what is meant by “natural balance, species restoration,”. Consider rewording both.
Response 6: This is reformulated as "urban tree restoration..."
Point 7: [Line 71] – Delete “of”.
Response 7: done
Point 8: [Line 137-138] – Change sentence from “This concerns the scientific names, families and genera of the species.on the one hand.” To “On the one hand, this concerns the scientific names, families and genera of the species.”
Response 8: This is done as recommended.
Point 9: [Lines 140 and 211] It says “DHP”, did you mean DBH? If so change, if not, define DHP.
Response 9: Yes we meant DBH and this is corrected as recommended.
Point 10: [Line 151] – Is it “Specific richness (S)” or “Species richness (S)”?
Response 10: It is Specific richness (S).
Point 11: [Lines 153-154] – Put “Shannon diversity index (H)” on the same line as “The Shannon diversity index varies both in terms of the number of species present” as you did with “Species richness (S)” on line 151.
Response 11: Done as recommended
Point 12: [Line 163-164] – Same comment as above, move “Pielou's equitability (Eq)” to the same line as it’s description.
Response 12: [Line 178] – Sentence begins with a spiral symbol, delete unless this was intentional.
Response 12: Done
Point 13: [Line 191] – Delete “()” from “(to obtain a percentage)”.
Response 12: Done
Point 13: [Line 199] – Sentence begins with a spiral symbol, delete unless this was intentional.
Response 13: this was intentional.
Point 14: [Line 213] – Change “Table II” to “Table 2”.
Response 14: This is done.
Point 15: [Line 216] – Change “(S=39);” to “(S=39),”.
Response 15: Done
Point 16: [Line 251] – The text on Figure 3 is blurry, replace with a higher resolution image.
Response 16: Done
Point 17: [Line 253-254] – Add comas and an “and” to the sentence “……the families of Anacardiaceae Euphorbiaceae Malvaceae Meliaceae Myrtaceae Rutaceae Sapotaceae.” It should then read “…..the families of Anacardiaceae, Euphorbiaceae, Malvaceae, Meliaceae, Myrtaceae, Rutaceae, and Sapotaceae.”
Response 17: This is done.
Point 18: [Line 283] – Graphs C, D, and E in Figure 4 all have species names that are on top of each other, making it difficult to read. Try to update the three graphs that the species are no longer depicted with the words on top of one another.
Response 18: For this, the issue is relating to the statistic valeur of species, that's why it is like this. But what is really interesting that the informations is well transcrit with a good narrative in the text (results).
Point 19: [Line 294] – Change sentence from “About of green areas and etablishments,….” To “Regarding the green areas and establishments,…..”.
Response 19: This is done. You can see it in the revised version of the manuscrip^t submitted.
Point 20: [Line 319-331] – Could this not also be the result of different urban forest management (i.e., planting, removals, etc.) across the various cities, varying levels of funding/staffing for urban forestry programs, potential natural disasters that may have impacted an area, local polies/ordinances, development codes, tree protection codes, mitigation efforts, landscape designs, etc. Thus, climatic conditions are just one of many variables that could affect species composition and diversity. It would be of value to make mention of this.
Response 20: Indeed. This is valuable and we use these suggestions by integrating them in the document.
Point 21: [Line 373] – Change “land-use” to “land use”.
Response 21: Done
Point 22: [Line 374] – delete the work “of”.
Response 22: done
Point 23: [Line 399] – Add “and” between “urban environment” and “biophysical conditions”.
Response 24: Done
Point 25: [Line 434] – Change “These concerns:-funding urban ecology research….” To “These include funding urban ecology research……..”
Response 23: Done as recommended.
Point 24: [Line 435] – Remove extra spaces between “tree species in the context”.
Response 24: It is done.
Point 25: [Line 436] – Change “-the factors……” to “the factors…..”.
Response 25: Done as recommended.
Point 26: [Line 573] – Delete “46.”.
Response 26: Done
We want to note that the reviewer 2 has made more contributions concerning spelling, conjugation and formatting errors that we valued very well. In addition, we have taken into account his suggestions concerning the limits of the study in the conclusion section.
Best regards.

Reviewer 3 Report (New Reviewer)
The article submitted for review concerns an important issue - the importance of trees / forest in shaping urban space in accordance with the assumptions of sustainable development. The authors correctly conducted research and analysis of the collected materials.
The reviewer suggests some additions or corrections:
1. Line 48: Can you quote the current urbanization index values?
2. Lines 83-88: the goals set, especially the last one (poems 87-88) according to the reviewer, have not been achieved. The study did not undertake research on the impact of climate change on urban greenery, and even more so did not provide recommendations for the protection of urban species against the effects of climate change.
3. Lines 104-107: I suggest citing climate data from lowest to highest value, e.g. "The average temperature in the coastal zone is 26.8°C with extremes of 16.5°C and 36.6°C."
The paper mainly presents floristic diversity in relation to spatial units. This is not a study on the city's adaptation to projected climate change. Therefore, the use of the term "climate adaptation" in the keywords is not justified.
Author Response
Dear Reviewer
Thank you for taking the time to provide quality comments on our manuscript despite your busy schedule.
Taking into account your comments, please find the answers point by point in the lines below.
Point by pont responses
Point 1: Line 48: Can you quote the current urbanization index values?
Response 1: The current urbanization index values are not provided in local documents.
Point 2: Lines 83-88: the goals set, especially the last one (poems 87-88) according to the reviewer, have not been achieved. The study did not undertake research on the impact of climate change on urban greenery, and even more so did not provide recommendations for the protection of urban species against the effects of climate change.
Response 2: The study promises in its objectives to prioritize the top species that have ecological importance in order to allow future study relating to climate impacts on them and their preservation. Thus, the impacts of climate change are relating to future researches from the results and recommendations the study did in the results and conclusion parts. As promised, the recommendations are reformulated, taking mainly account suggestions to conduct climate impacts on urban trees. You can see them in the revised version of the manuscript.
Point 3: Lines 104-107: I suggest citing climate data from lowest to highest value, e.g. "The average temperature in the coastal zone is 26.8°C with extremes of 16.5°C and 36.6°C."
Response 3: This is done as recommended. You can see it in the revised version.
Point 4: The paper mainly presents floristic diversity in relation to spatial units. This is not a study on the city's adaptation to projected climate change. Therefore, the use of the term "climate adaptation" in the keywords is not justified.
Response 4: This is right but we were thinking about the ecosystem services obviousely provided these trees to adapt citizens to climate change because this was the subject of our previeus paper published in MDPI Sustainability. But as recommended, we canceled from the key words.
Best regards.

Round 2
Reviewer 1 Report (New Reviewer)
Dear Editor,
I think that the manuscript is improved in the present version. I accept that it could be published in Climate Journal.
Best regards,
Bouchra El Omari
Author Response
This evaluator agree with the responses we provided.
Best regards
This manuscript is a resubmission of an earlier submission. The following is a list of the peer review reports and author responses from that submission.
Round 1
Reviewer 1 Report
Trees diversity and species with high ecological importance for 1 resilient urban area: evidence from Cotonou city in West Africa
Summary: This manuscript provides a summary of vegetative diversity across the city of Cotonou, Benin, West Africa. The authors use several metrics to quantify species richness and diversity, and to determine how they vary across six different land use types in the city.
There is merit to the paper, and the findings may be of use to urban planners across Benin and West Africa who intend to increase canopy cover for climate mitigation efforts. However, there are many areas in which this manuscript must be improved. Most notably, the methods section is not strong. There are descriptions of some of the metrics that will benefit revision. Additionally, there are some areas in which the metrics are not referred to properly, or are referred to using differing names and symbols. The size of study plots varied across the study area, but no justification was provided as to why. Additionally, while there are 13 administrative districts within the city, the research team only sampled from 8. Again, there was no description as to why. This makes me very concerned about the validity of the findings that are reported within.
Comments
Abstract:
Consider adding brief text that explains that you also accounted for IVI in the study, why, and that this was used to contextualize species abundance.
Consider adding text that explains if there were differences in species found among the different land use types.
Line 14: Instead of “Here, we took Cotonou city to do that.” perhaps rephrase to highlight that this will be one of the premiere studies to investigate climate impacts on trees in an urbanized part of Benin, and perhaps provide brief contextual reference for the city’s size or prominence relative to other cities in Benin.
Introduction
Lines 46-48: Sentence should be reworded for clarity. Perhaps rather than beginning with “while”, consider something along the lines of “Simultaneously”.
Lines 49-51: Additional benefits should be added, including urban heat island mitigation, stormwater mitigation, and known positive outcomes on human physical and mental health.
Methodology
Lines 87-88: Add the date for the census in the text.
Lines 93-94: Be sure to note if any streams, canals, or other surface water bodies connect the lake and the ocean.
Lines 102-104: Provide clarification and justification you decided to sample in 8 rather than all 13 districts. This seems somewhat arbitrary without supporting text to justify.
Lines 114-116: Why were two different plot sizes used? This will need to be explained. I am concerned that the use of two different plot sizes will invalidate the comparisons upon which this study is based.
Line 119: Should say DBH rather than Dhp
Lines 116-120: This needs better clarity. Measuring DBH is part of baseline data collection when conducting a stand inventory. What do the authors mean when they mention that they conducted a standard inventory? It would be better to state that inventory data were collected in each plot, and then to list the data that were collected.
Lines 122-124: This section needs to be revised as somethings are not written correctly. In these lines, two metrics are denoted with an S, and Pielou metric is referred to by one name here that it is not referred to later on in the text. Nowhere in the text is it referred to by its proper name: Pielou’s evenness index.
Lines 132-134: Please revise as parts of this text are not fully clear. For example, what is meant by “it is expressed in bits?”
Lines 142-147: Explanation of Pielou’s index must be revised for clarity. In one part of the text, it is referred to as E, then in another part it is referred to as S. There is never a full equation provided to determine how it is calculated.
Lines 148-149: Provide some text-based description of the difference between effective diversity and theoretical maximum diversity.
Lines 150-155: This text is somewhat unclear. What does S in the above equation represent? How is it calculated?
Lines 181-183: This is too vague. Please provide a more detailed description of the analyses that were conducted. Additionally, please consider reporting descriptive statistics with your findings.
Results
Line 187: I think this should be DBH rather than Dhp.
186-195: This text is challenging to read as it is not clear what S denotes. Should this be n instead?
Lines 212-213: Clarification is needed here. What is meant by the “the distribution of trees in regular?”
Figure 3: Should say Genus rather than Gender.
Lines 235-237: Clarification is needed here. What do the numbers in parentheses indicate? If it is the number of plots where they are present, please denote with “n =” and consider indicating what percentage of plots within each land use type for which these species were present.
Figure 4 must be improved. Is there a unit with which abundance is being measured (perhaps %)? If so, please indicate that on the appropriate axis label. Also, please revise the figure caption. A description should be provided here for the data that is displayed in each of the respective panels of the figure.
Lines 261-262: This explanation of the IVI must be improved for correctness and clarity. Use of the word importance is vague, subjective to varied interpretations, and not fully correct. This metric provides a means by which characterizing species dominance. Please be sure to be consistent with a description that communicates this here and in other parts of the text.
Lines 283-284: Please revise the figure caption to include a description of what each of the acronyms denotes.
Discussion
Lines 290-291: Rewording is needed here. As you are comparing species richness across varying cities, I don’t think the goal is to reinforce observations from other locations in which there are completely different environmental, social, and land-use related conditions. It would be better, instead, to say that your findings are within the range of those in other cities within West Africa.
Line 346: Add period after brackets.
Lines 351-352: The use of “storming” in this context seems out of place and colloquial. Please use terminology that is more aligned with the field of (urban) forest ecology.
Line 369: Word choice – I’m not sure the potential is the best word here.
There are some areas where word choice can be improved.
Author Response
Dear reviewer
We'd like to thank you very much for taking the time to read our manuscript and for your pertinent comments on the quality of this work, despite your busy schedule.
That said, please find below the responses to your comments and their consideration in the final version.
RESPONSES TO COMMENTS:
About the abstract
Consider adding brief text that explains that you also accounted for IVI in the study, why, and that this was used to contextualize species abundance.
Consider adding text that explains if there were differences in species found among the different land use types.
Response: This is done and integrated in the final version of the manuscript resubmitted.
Line 14: Instead of “Here, we took Cotonou city to do that.” perhaps rephrase to highlight that this will be one of the premiere studies to investigate climate impacts on trees in an urbanized part of Benin, and perhaps provide brief contextual reference for the city’s size or prominence relative to other cities in Benin.
Response: This is taken into account in the finale version of the manuscript.
ABOUT THE INTRODUCTION
Lines 46-48: Sentence should be reworded for clarity. Perhaps rather than beginning with “while”, consider something along the lines of “Simultaneously”.
Response: We dealed with this in the introduction section.
Lines 49-51: Additional benefits should be added, including urban heat island mitigation, stormwater mitigation, and known positive outcomes on human physical and mental health.
Response: We added that: Atchadé et al. (2023) have demonstrated through stakeholders’ consultation, that urban trees in Cotonou city have the ability mitigate urban heat island, stormwater and strengthen human health.
ABOUT THE METHODOGY
Lines 87-88: Add the date for the census in the text.
Response: This is done in the last version (We have to mention that The fieldwork ran from May 2022 to October 2022.)
Lines 93-94: Be sure to note if any streams, canals, or other surface water bodies connect the lake and the ocean.
Response: Nothing about this in the study area.
Lines 102-104: Provide clarification and justification you decided to sample in 8 rather than all 13 districts. This seems somewhat arbitrary without supporting text to justify.
Response: With the means available and the time available for data collection, we randomly selected 8 of the 13 boroughs for work exchange. This sampling approach follows the scientific logic used by Moussa et al. (2020) and Foléga et al. (2019) respectively in the cities of Niamey (Niger) and Atakpamé (Togo). These countries all border Benin.
Lines 114-116: Why were two different plot sizes used? This will need to be explained. I am concerned that the use of two different plot sizes will invalidate the comparisons upon which this study is based.
Response: This approach relating to plots sizes follows what recommended by Thiombiano at al. (2016) to hamonize forest sampling in West Africa area. As the shape of the roadway is purely different from that of other occupancy units such as administrative areas and the like, rectangular plots were installed along the tracks.
Line 119: Should say DBH rather than Dhp
Response: This is corrected in the new manuscript version.
Lines 116-120: This needs better clarity. Measuring DBH is part of baseline data collection when conducting a stand inventory. What do the authors mean when they mention that they conducted a standard inventory? It would be better to state that inventory data were collected in each plot, and then to list the data that were collected.
Lines 122-124: This section needs to be revised as somethings are not written correctly. In these lines, two metrics are denoted with an S, and Pielou metric is referred to by one name here that it is not referred to later on in the text. Nowhere in the text is it referred to by its proper name: Pielou’s evenness index.
Response: floristic surveys were carried out. This concerns the scientific names, families and genera of the species. (this is mentioned in the new version of the manuscript)
Lines 132-134: Please revise as parts of this text are not fully clear. For example, what is meant by “it is expressed in bits?”
Response: This means "bits is the scientific unit of this index". We have corrected it in the new version of the manuscript.
Lines 142-147: Explanation of Pielou’s index must be revised for clarity. In one part of the text, it is referred to as E, then in another part it is referred to as S. There is never a full equation provided to determine how it is calculated.
Response:
We have corrected the confusion between the equitability index noted E in some places and S in others.
But for the fiormula, equation 2 is the calculation formula used to estimate this index. Equitability reflects the degree of diversity achieved by a stand, and corresponds to the ratio between the effective diversity (H) and the theoretical maximum diversity (Hmax) (equation 2).
Lines 148-149: Provide some text-based description of the difference between effective diversity and theoretical maximum diversity.
Response: Effective diversity is H = - ∑pi log2 pi; (in equation 1) where
Pi (between 0 and 1): relative proportion of the number of individuals of species i in the total number of individuals of all species concerned; Pi = ni /∑ ni; with ni: number of individuals of species i and ∑ ni: total number of individuals of all species. On the other hand, H is maximum (Hmax in this context) when species are identically abundant.
Response: This is done in the text (last version of the maunscript).
Lines 150-155: This text is somewhat unclear. What does S in the above equation represent? How is it calculated?
Response: The letter S in this section stands for Species Richness (S), as mentioned at the beginning of the TAXONOMIC DIVERSITY section. This was the very first index defined. Specific richness (S): the number of species present in each zone of the city of Cotonou). In this study, it is represented on each of the city's land-use units. We've also corrected the mistake of replacing it with the letter R in one place.
Lines 181-183: This is too vague. Please provide a more detailed description of the analyses that were conducted. Additionally, please consider reporting descriptive statistics with your findings.
Response: In reality, the VEGAN package is used to calculate the various ecological diversity indices describing ecosystems in R software. In this package cited in bibliographic references, the various formulas described above are integrated.
ABOUT THE RESULTS
Line 187: I think this should be DBH rather than Dhp.
Response: Corrected
186-195: This text is challenging to read as it is not clear what S denotes. Should this be n instead?
Response: S stands for Species Richness in each land use units, as described above in the methodology section.
Lines 212-213: Clarification is needed here. What is meant by the “the distribution of trees in regular?”
Response: This expressees the eveness of species. It corrected in the text.
Figure 3: Should say Genus rather than Gender.
Response: This is corrected in the text.
Lines 235-237: Clarification is needed here. What do the numbers in parentheses indicate? If it is the number of plots where they are present, please denote with “n =” and consider indicating what percentage of plots within each land use type for which these species were present.
Response: Pls, it is not the number of plots but the number of individuals (trees of the species in each land use type) according to the rank/abundance curve of the figure 4. We emphazie that this is the top 5 species present in each land use type in the city of the Cotonou.
Figure 4 must be improved. Is there a unit with which abundance is being measured (perhaps %)? If so, please indicate that on the appropriate axis label. Also, please revise the figure caption. A description should be provided here for the data that is displayed in each of the respective panels of the figure.
Response: In the Vegan package used and referenced, the units (in Abundance and species rank) are default. And we have corrected (in text) that the abscissa axis (species rank) expresses the top 5 species of the occupancy unit, while the ordinate axis expresses the abundance (in number of trees) of individuals of each species.
Lines 261-262: This explanation of the IVI must be improved for correctness and clarity. Use of the word importance is vague, subjective to varied interpretations, and not fully correct. This metric provides a means by which characterizing species dominance. Please be sure to be consistent with a description that communicates this here and in other parts of the text.
Lines 283-284: Please revise the figure caption to include a description of what each of the acronyms denotes.
Response: It is done like recommended in the new manuscript.
ABOUT THE DISCUSSSION
Lines 290-291: Rewording is needed here. As you are comparing species richness across varying cities, I don’t think the goal is to reinforce observations from other locations in which there are completely different environmental, social, and land-use related conditions. It would be better, instead, to say that your findings are within the range of those in other cities within West Africa.
Response:This is done in the text.
Line 346: Add period after brackets.
Response: done
Lines 351-352: The use of “storming” in this context seems out of place and colloquial. Please use terminology that is more aligned with the field of (urban) forest ecology.
Response: done with "invasion"
Line 369: Word choice – I’m not sure the potential is the best word here.
Response: changed with "ability"
Thank very much for your support.
Sincerely
Reviewer 2 Report
It is a good work, which is complete and provides valuable information.
To improve the present version, I recommend increasing the introduction.
I have no problem with the methodology, results and discussion sections.
Standardize lines 148-149 according to format
Author Response
Dear reviewer
We'd like to thank you very much for taking the time to read our manuscript and for your pertinent comments on the quality of this work, despite your busy agenda.
We take into account of your observations in the introductive part of the new version. You can see this within.
Sincerely
Reviewer 3 Report
This manuscript investigated the diversity of tree species within an urban area. However, it is imperative that substantial revisions be made. It is better to thoroughly review your writing, paying particular attention to the consistent use of tense. The manuscript, in its present form, lacks the required data to substantiate the conclusions. Furthermore, it's important to recognize that tree diversity encompasses a broad spectrum of aspects, not limited to species diversity alone. It also includes diversity in terms of size and age, but the current inventory fails to account for due to its simplistic approach. Another notable question is the absence of any analysis pertaining to the influence of climate on tree diversity. Given that the subject matter revolves around trees, it is highly recommended that you consider submitting your manuscript to a journal specializing in forestry, as this would be a more fitting platform.
Comments:
1. Line 12: ancient trees – old trees. Why only ancient trees? I have not seen this term anymore.
2. Line 14: Here, we took Cotonou city to do that. – Revise this wired expression.
3. Line 15: plant diversity – tree diversity
4. Line 15: city’s land-use units – what do you mean?
5. Line 57: According to [16,9] – wrong expression
6. Line 62: the work of [18] – wrong expression. Please revise this issue throughout the manuscript.
7. Line 73 – 75: The project's objective lacks clarity; elucidate your scientific question and formulate a precise hypothesis. Provide more information about the diversity (e.g., composition or structure?)
8. Figure 1. It is too hard to identify these plots. Replace the figure with a high resolution figure.
9. Line 119: Why only trees larger than 10 cm? This inventory ignored a lot of saplings.
10. Figure 4. Texts overlaid.
11. Line 362: I believe you are suggesting that management practices can exert an influence on the tree species present in an urban area. Consequently, both management and planting strategies can significantly impact not only the diversity of tree species but also the structural diversity within the area. As such, without considering the implications of management and planting on diversity, this analysis lacks applicability in informing policy development.
12. Discussion: Urban forests are typically subject to rigorous management. The abundance of specific tree species can be considerably influenced by planting strategies. However, I have not come across any information addressing this particular issue in the current manuscript.
13. Discussion: All the plots under study are situated within a single city, and it seems unlikely that there would be significant variations in climate conditions across these plots. Consequently, drawing conclusions regarding the influence of climate on tree diversity from such a limited geographical scope appears questionable. Furthermore, the inclusion of discussions pertaining to climate in the conclusion section does not seem justified, given the lack of substantial climate variability within the study area. It would be better to exclude these discussions.
It is better to thoroughly review your writing, paying particular attention to the consistent use of tense.
Author Response
Dear reviewer
We'd like to thank you very much for taking the time to read our manuscript and for your pertinent comments on the quality of this work, despite your busy schedule.
Please, see below our responses
- Line 12: ancient trees – old trees. Why only ancient trees? I have not seen this term anymore
Response: done in the new version.
2. Line 14: Here, we took Cotonou city to do that. – Revise this wired expression.
Response: This is done
3. Line 15: plant diversity – tree diversity
Response: done
4. Line 15: city’s land-use units – what do you mean?
Response: According to Folega et all. (2019) and Moussa et al. (2020) , land use unit means type of the land use, compartment of urban area.
5. Line 57: According to [16,9] – wrong expression
Response: This is corrected in the text.
6. Line 62: the work of [18] – wrong expression. Please revise this issue throughout the manuscript.
Response: this is done.
7. Line 73 – 75: The project's objective lacks clarity; elucidate your scientific question and formulate a precise hypothesis. Provide more information about the diversity (e.g., composition or structure?)
Response: This is done as follows:
the present work aims to fill these knowledge gaps by:
- evaluating the floristic diversity in term of composition of the different land use units through different ecological indexes.
- prioritizing the top species that have ecological importance in order to allow future study relating to climate impacts on them and their preservation.
-making recommendations to protect urban species from climate change impacts and other threats (like species invasion).
The results could help in guiding the green development policies of future African and Beninese cities.
9. Line 119: Why only trees larger than 10 cm? This inventory ignored a lot of saplings.
Response: This work was oriented in this sens. It was schedule to evaluate older trees contributions to climate resilience and mitigation in the city of Cotonou. The manuscript is a part of the results from the study. Another aprt of the work is already published. In the original protocol, we have set to explore urban trees and woody abitiy to contribute to climate relisience and mitigation.
10. Figure 4. Texts overlaid.
Response: Indeed, you are right but this is due to the fact that some of these top 5 abundant species have the same number in the type of the land use. But all the informations in the figure were transcrited into the text, and this make them available for understanding.
11. Line 362: I believe you are suggesting that management practices can exert an influence on the tree species present in an urban area. Consequently, both management and planting strategies can significantly impact not only the diversity of tree species but also the structural diversity within the area. As such, without considering the implications of management and planting on diversity, this analysis lacks applicability in informing policy development.
Response: this is done in the conclusion with new recommendations.
12. Discussion: Urban forests are typically subject to rigorous management. The abundance of specific tree species can be considerably influenced by planting strategies. However, I have not come across any information addressing this particular issue in the current manuscript.
Response: This is done in the conclusion of the new version
13. Discussion: All the plots under study are situated within a single city, and it seems unlikely that there would be significant variations in climate conditions across these plots. Consequently, drawing conclusions regarding the influence of climate on tree diversity from such a limited geographical scope appears questionable. Furthermore, the inclusion of discussions pertaining to climate in the conclusion section does not seem justified, given the lack of substantial climate variability within the study area. It would be better to exclude these discussions.
Response: Setting study objectives with some features relating to climate change in the end of the introduction makes this question treatable. This is done in the new version.
Thank you very much for your contribution. We really valued your comments and onservstions.
Sincerely